# *STAG2* mutations in the normal colon induce upregulation of oncogenic pathways in neighbouring wildtype cells

Wei Ni Yew [ID][1], Christopher James Dean[2], Dedrick Kok Hong Chan[ID][1,3,4]*

1 Department of Surgery, Yong Loo Lin School of Medicine, National University of Singapore, Singapore, Singapore, 2 Genomics and Data Analytics Core, Cancer Science Institute, Singapore, Singapore, 3 Division of Colorectal Surgery, University Surgical Cluster, National University Hospital, Singapore, Singapore, 4 NUS Centre for Cancer Research, Yong Loo Lin School of Medicine, National University of Singapore, Singapore, Singapore

* surckhd@nus.edu.sg

## Abstract

While driver mutations in the normal colon have been described, characterizing the role and function of these driver mutations in relation to colorectal oncogenesis remains incomplete. Here, we investigated the role of *STAG2* mutants in the normal colon using patient-derived wildtype organoids. Using CRISPR-Cas9 gene editing, we generated *STAG2* mutants, and co-cultured these mutants with wildtype organoids, mimicking the presence of such *STAG2* mutants in the normal colon. We sought to determine the transcriptional impact of co-culture using scRNAseq. Surprisingly, we uncovered a possible cell-cell interaction between *STAG2* mutants and wildtype organoids, in which wildtype organoids in co-culture with *STAG2* mutants upregulated known oncogenic pathways. This included the upregulation of TNFα-signaling, as well as KRAS-signaling in wildtype organoids. These results suggested that *STAG2* mutant cells exert a pro-oncogenic effect in a cell interactive manner, instead of via a cell autonomous approach. In conclusion, our findings demonstrate a novel mechanism of colorectal oncogenesis which can support further investigation.

## Introduction

There has been recent interest in the role of somatic driver mutations in phenotypically normal organs, particularly in relation to oncogenesis [1]. In the human oesophagus, *NOTCH1* mutations were found to drive clonal expansion, outcompeting early tumours, and therefore impeding the onset on oesophageal cancer [2,3]. In the colon, whole genome sequencing of single crypts revealed the presence of somatic mutations including *AXIN2*, *FBXW7*, and *STAG2* [4]. Our earlier work on *FBXW7* mutations in the normal colon showed that such mutations repress the impact of a

**Data availability statement:** The datasets generated and/or analysed during the current study are available in the NCBI Sequence Read Archive with the accession number PRJNA1227013, and may be accessed from https://www.ncbi.nlm.nih.gov/sra/PRJNA1227013.

**Funding:** DKHC was supported by an N2CR-CSDU CS Pilot Grant as well as an NUHS Clinician Scientist Program grant (NCSP2.0/2023/NUHS/DCKH).

**Competing interests:** The authors have declared that no competing interests exist.

subsequent *APC* transcriptional response, therefore preventing the initiation of colorectal cancer (CRC) [5].

We sought to determine the impact of *STAG2* mutations in the normal colon on oncogenesis. STAG2 is a critical component of the cohesin complex together with three other core units, SMC1A, SMC3 and RAD21. Together, the cohesion complex forms a ring-shaped structure which encircles chromatin during the early G1 phase of the cell cycle, and concatenates sister chromatids during the subsequent S phase [6]. Within the cohesin complex, STAG2 may be interchanged with STAG1. STAG2 is essential for chromatin cohesion at centromeres and along chromosome arms, while STAG1 is essential for chromatin cohesion at telomeres [7,8]. More recently, STAG2 has also been found to play a critical role in shaping the chromatin architecture, in some cases by altering enhancer-promoter interactions, and in others through the formation of chromatin loops. These architectural changes result in the differential expression of genes, and may result in changes in cellular differentiation which lead to cancer. Through such alterations in chromatin architecture, associations between *STAG2* mutations and acute myeloid leukemia [9], glioblastoma multiforme [10], as well as bladder cancer [11] have been found. *STAG2* mutations have however not been studied within the context of CRC.

In this study on the role of *STAG2* mutations in the normal colon, we knocked out the *STAG2* gene using CRISPR-Cas9 gene editing. Co-culture of *STAG2* mutants and wildtype allowed us to determine effects of cell-cell interaction which may contribute towards tumour initiation. Single-cell RNA sequencing (scRNAseq) analysis was performed which revealed the effect of the *STAG2* mutant gene on neighbouring wildtype cells. Our findings suggest that *STAG2* mutants in the normal colon may subtly upregulate carcinogenic pathways such as TNFα signaling in neighbouring wildtype cells, providing a hitherto undescribed novel mechanism of colorectal carcinogenesis.

## Results

### Generation of *STAG2* mutant organoids

Patient derived adult stem cell organoids were generated from the normal tissue of patients undergoing colectomy (Fig 1a). Details regarding the gender, age and site of colon tissue harvested can be found in Table 1. We used CRISPR-Cas9 gene editing to introduce knockout mutations of the *STAG2* gene using our previously described protocol [12]. There were no gross phenotypic differences between *STAG2* wildtype and mutant organoids (Fig 1b). Validation of *STAG2* knockout was thus confirmed using western blot (Fig 1c), as well as immunofluorescence with anti-STAG2 antibody (Fig 1d).

### *STAG2* mutant organoids exhibit increased crypt proliferation in vitro

Previously, *STAG2* mutant stem cells in the colonic crypt visualized on formalin-fixed paraffin-embedded (FFPE) slides were found to exhibit increased cellular proliferation, outcompeting neighbouring wildtype stem cells, leading to accelerated fixation within the crypt and subsequent clonal expansion of *STAG2* mutant crypts [13]. We sought to determine whether a similar phenotype could be observed with organoids *in vitro*. Generated organoids were cultured in human differentiation media (HDM).

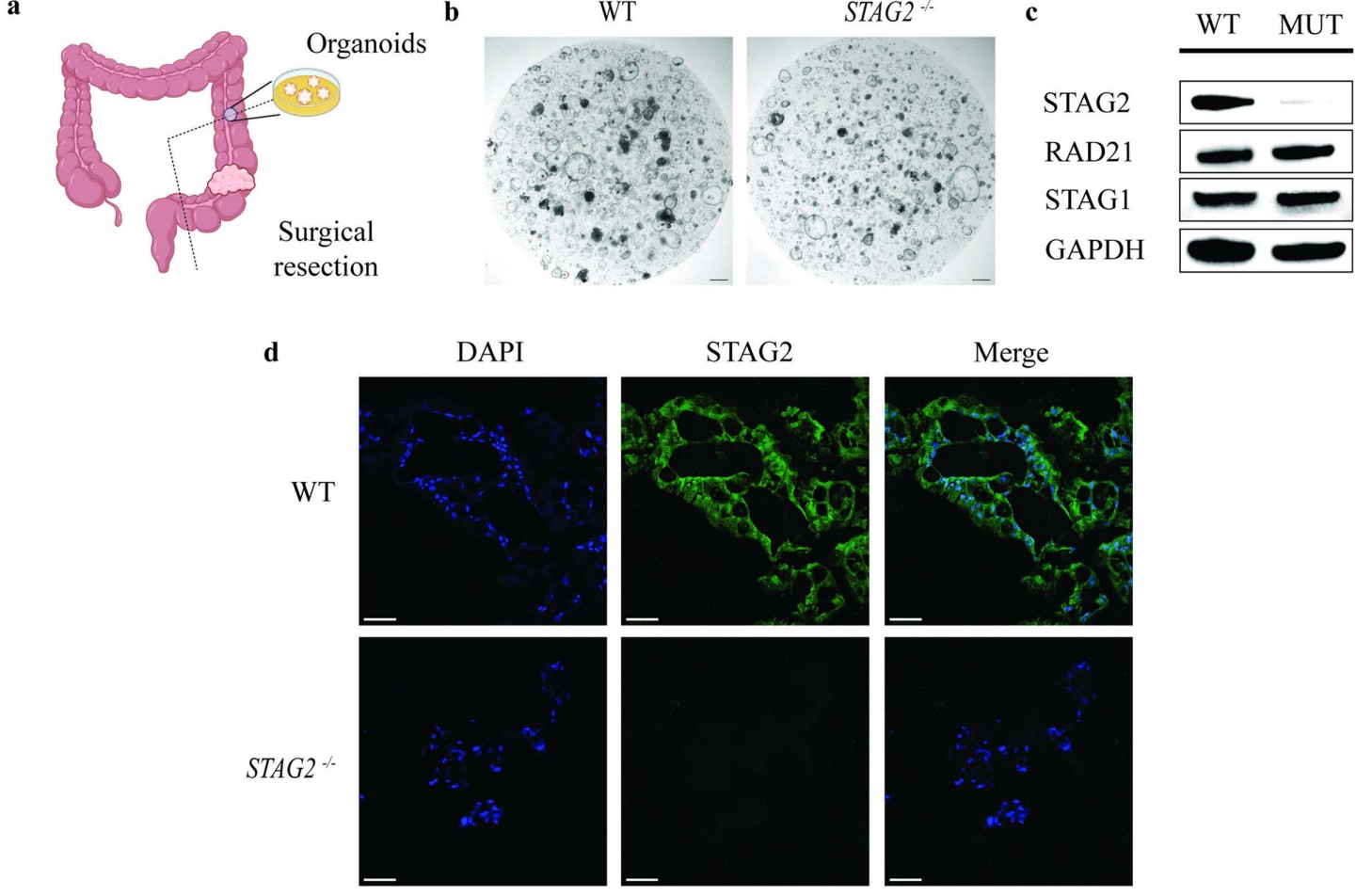

**Fig 1. Generation of patient-derived organoids. a.** Colonic organoids are generated from the phenotypically normal resection margin of surgical specimens. **b.** Brightfield microscopy of wildtype and *STAG2* mutants displayed no observable phenotypic differences. Scale bars represent 500µm. **c.** Western blot validation showing loss of STAG2 protein in *STAG2* mutant knockout organoids, with preservation of RAD21 and STAG1, the other components of the cohesion complex. **d.** Immunofluorescence of *STAG2* wildtype and mutant organoids with DAPI nuclear stain (blue) and STAG2 (green). Scale bars represent 50µm. All experiments were performed with N = 3 biological replicates.

*APC* mutant organoids were used as a positive control for increased cellular proliferation and crypt formation. Organoids grown in differentiation media rapidly lost their cystic appearance, becoming thick-walled, developing crypt-like protuberances from the organoid centre which resulted in irregular shapes and reduced circularity (Fig 2a). We therefore assessed individual crypts for circularity as well as the density of crypts/mm2 in wildtype, *STAG2* mutant, and *APC* mutant organoids. We observed decreased circularity and increased crypt formation per mm2 in APC organoids as expected (Fig 2b and 2c). In *STAG2* mutant organoids, we also observed increased crypt formation concordant with colonic crypts on FFPE slides (Fig 2b and 2c). These findings confirmed that *STAG2* mutant organoids recapitulate proliferation characteristics observed *in vivo*, and therefore may be used to infer its role in normal tissue.

### *STAG2* mutant organoids interact with neighbouring wildtype cells to induce carcinogenic pathways

Data from The Cancer Genome Atlas (COAD-READ) demonstrates that the incidence of *STAG2* mutations in CRC is only 2.7% [14]. This suggests that although *STAG2* mutants demonstrate a clonal advantage in normal colonic epithelia,

**Table 1. Table demonstrating the age, gender, and site of colon harvested for specimens used in the generation of organoids.**

| Gender | Age | Site of colon harvested |
|--------|-----|------------------------|
| F | 61 | Rectosigmoid Colon |
| M | 86 | Ascending Colon |
| F | 60 | Rectosigmoid Colon |
| M | 66 | Rectosigmoid Colon |
| F | 67 | Sigmoid Colon |
| M | 75 | Rectosigmoid Colon |
| M | 70 | Rectosigmoid Colon |
| M | 82 | Rectosigmoid Colon |
| M | 68 | Rectosigmoid Colon |
| M | 65 | Descending Colon |
| M | 62 | Ascending Colon |
| M | 71 | Descending Colon |
| F | 63 | Rectosigmoid Colon |
| M | 57 | Rectosigmoid Colon |
| M | 50 | Descending Colon |
| M | 54 | Sigmoid Colon |
| M | 70 | Sigmoid Colon |
| M | 69 | Ascending Colon |
| F | 71 | Ascending Colon |
| M | 72 | Ascending Colon |

*STAG2* mutant cells are unlikely to initiate cancer via a cell autonomous effect. *STAG2* mutant cells may instead exert a cell interactive effect with neighbouring wildtype cells. To study this potential cell interactive effect, we co-cultured *STAG2* mutant cells with wildtype cells, and performed single-cell RNA sequencing (scRNAseq) on the co-cultured cells.

Co-cultured cells on scRNAseq clustered into four clusters (Fig 3a). There was a higher proportion of *STAG2* mutant cells in Cluster 3, while there was a higher proportion of wildtype cells in Clusters 0 and 2 (Fig 3b–3d). Cluster 1 appeared to have equal proportions of *STAG2* mutant and wildtype cells.

The top 20 genes differentiating *STAG2* wildtype and mutant cells are represented in Fig 4a. Unsurprisingly, *STAG2* expression exhibited the greatest difference between wildtype and mutant clusters. Among these 20 genes, only *GPS2*, *SPTSSA*, *H4C3*, and *HNRNPA0* had average log2FC greater than 1, suggesting that differences in the expression of the remaining genes, while statistically significant, remained subtle. Expression of these genes was increased in wildtype cells relative to *STAG2* mutants. *G Protein Suppressor 2* (*GPS2*) is a tumour suppressor gene which inhibits PI3K/AKT-mediated cellular proliferation and tumour growth in breast cancer cell lines [15]. *Serine palmitoyltransferase small subunit A* (*SPTSSA*) encodes a protein which catalyses the rate-limiting step in sphingolipid biosynthesis [16]. The role of sphingolipid biosynthesis in cancer remains incompletely characterized. While increased *SPTSSA* expression was associated with oxidative stress and poorer survival in glioblastoma [17], increased sphingolipid production was instead associated with improved survival in renal cell carcinoma patients [18]. *H4 clustered histone 3* (*H4C3*) is a gene in which missense variants are associated with developmental delay and intellectual disability [19]. *H4C3* downregulation was also associated with pancreatic cancer in a recent study [20]. Finally, *heterogeneous nuclear ribonucleoprotein A0* (*HNRNPA0*) encodes an RNA-binding protein which complexes with heterogeneous nuclear RNA, and has been associated with a range of cancers. Mutations in the *HNRNPA0* gene have been associated with increased PI3K and ERK/MAPK signaling [21].

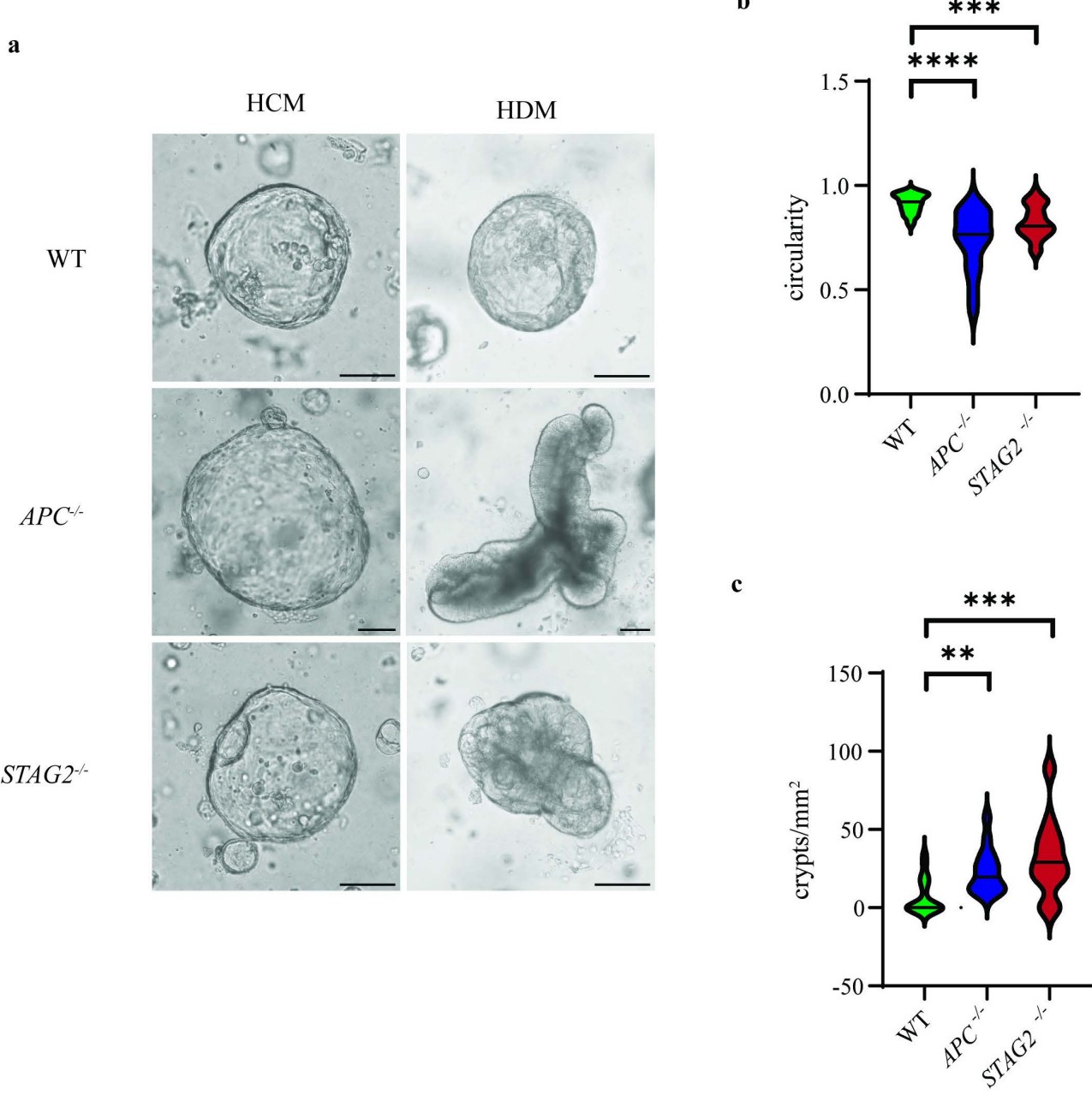

**Fig 2. *STAG2* mutant organoids display characteristics suggestive of increased crypt proliferation when cultured in differentiation media. a.** Brightfield images of organoids grown in human conditioned media (HCM) and then in human differentiation media (HDM). Expectedly, *APC* mutant organoids exhibited decreased circularity (b) and increased crypt proliferation **(c)**. In a similar way, *STAG2* mutant organoids also exhibited significantly decreased circularity (b) and increased crypt proliferation (c) compared with *STAG2* wildtype organoids. Scale bars are 100μm. All experiments were performed with N = 3 biological replicates.

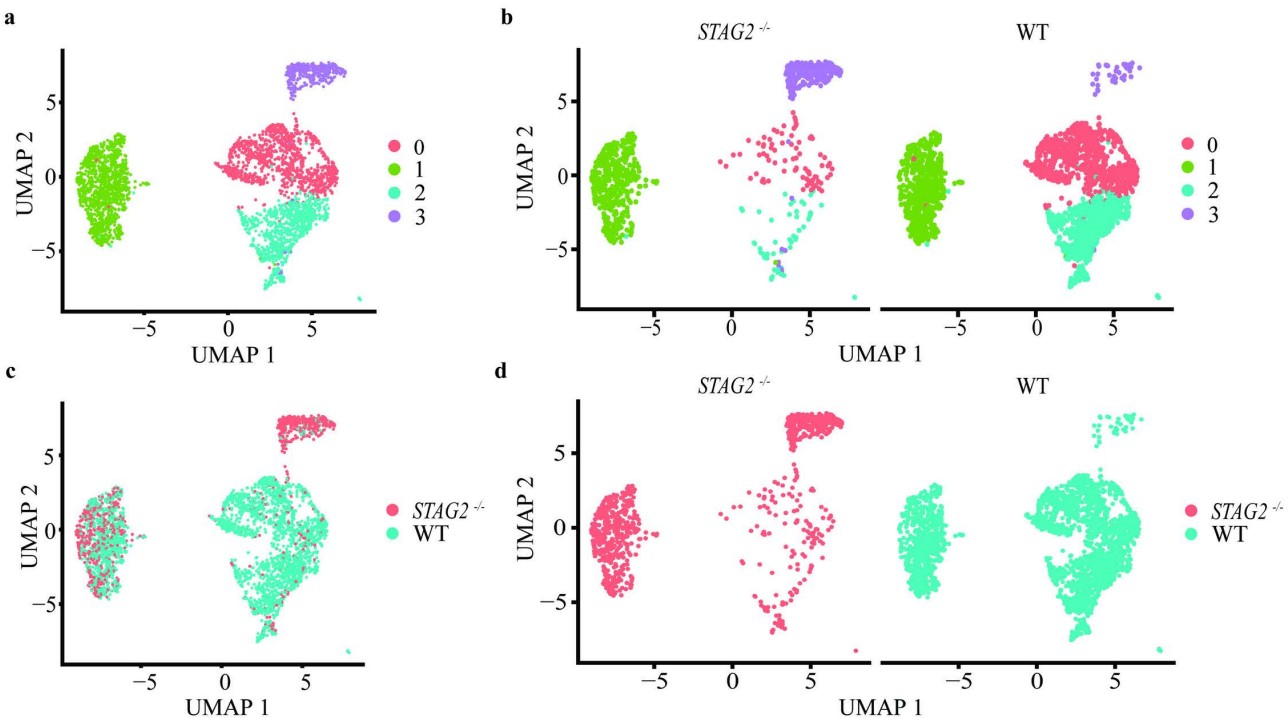

**Fig 3. ScRNAseq analysis of co-cultured *STAG2* wildtype and mutant organoids. a.** Analysed using Uniform Manifold Approximation and Projection (UMAP), co-cultured organoids clustered into four distinct groups. **b.** *STAG2* wildtype cells formed the majority in Clusters 0 and 2, while *STAG2* mutant cells formed the majority in Cluster 3. Panels (c) and (d) represent the above data with each point labelled as either *STAG2* wildtype (blue) or mutant (red).

Analysis of *STAG2* wildtype and mutant cells using Gene Set Enrichment Analysis (GSEA) for "Hallmark" gene sets (50 genes sets) revealed that only the "HALLMARK_TNFA_SIGNALING_VIA_NFKB" gene set was upregulated in wildtype compared to mutant at an FDR of <25% (Fig 4b). Intriguingly, no gene sets were upregulated in *STAG2* mutant relative to wildtype, lending further evidence that the primary effect of the *STAG2* mutant is on neighbouring wildtype cells.

Given our findings, we sought to confirm that co-cultured wildtype cells exhibited increased tumorigenicity using immunofluorescence staining. KI67 is a common marker used to assess cellular proliferation, On immunofluorescent staining, we observed an upregulation of fluorescent intensity in co-cultured wildtype organoids relative to *STAG2* mutant organoids ($p = 0.0089$) (Fig 4c and 4d). Likewise, Cyclin D1 is a key regulator of the cell cycle and is associated with tumour development and progression. Here, we also observed an upregulation of fluorescent intensity in co-cultured wildtype organoids relative to *STAG2* mutant organoids ($p = 0.0015$) (Fig 4e and 4f).

## Oncogenic pathways are upregulated in scRNAseq clusters with predominantly wildtype cells

Further analysis was undertaken using Gene Set Enrichment Analysis (GSEA) for "Hallmark" gene sets (50 genes sets) for individual clusters. A heatmap of differentially expressed genes by cluster is found in S1 Fig. Cluster 0 and 2 were predominantly comprised of cells from the *STAG2* wildtype population. We observed that the "HALLMARK_TNFA_SIGNALING_VIA_NFKB" which had been upregulated in wildtype cells was also upregulated in both clusters with FDR < 25% and nominal *p*-value < 5% (Fig 5a and 5b). In Cluster 0, other gene sets which were upregulated included "HALLMARK_ ANGIOGENESIS" (Fig 5c) and "HALLMARK_EPITHELIAL_MESENCHYMAL_TRANSITION" (Fig 5d). In Cluster 2, other gene sets which were upregulated included "HALLMARK_EPITHELIAL_MESENCHYMAL_TRANSITION" (Fig 5e),

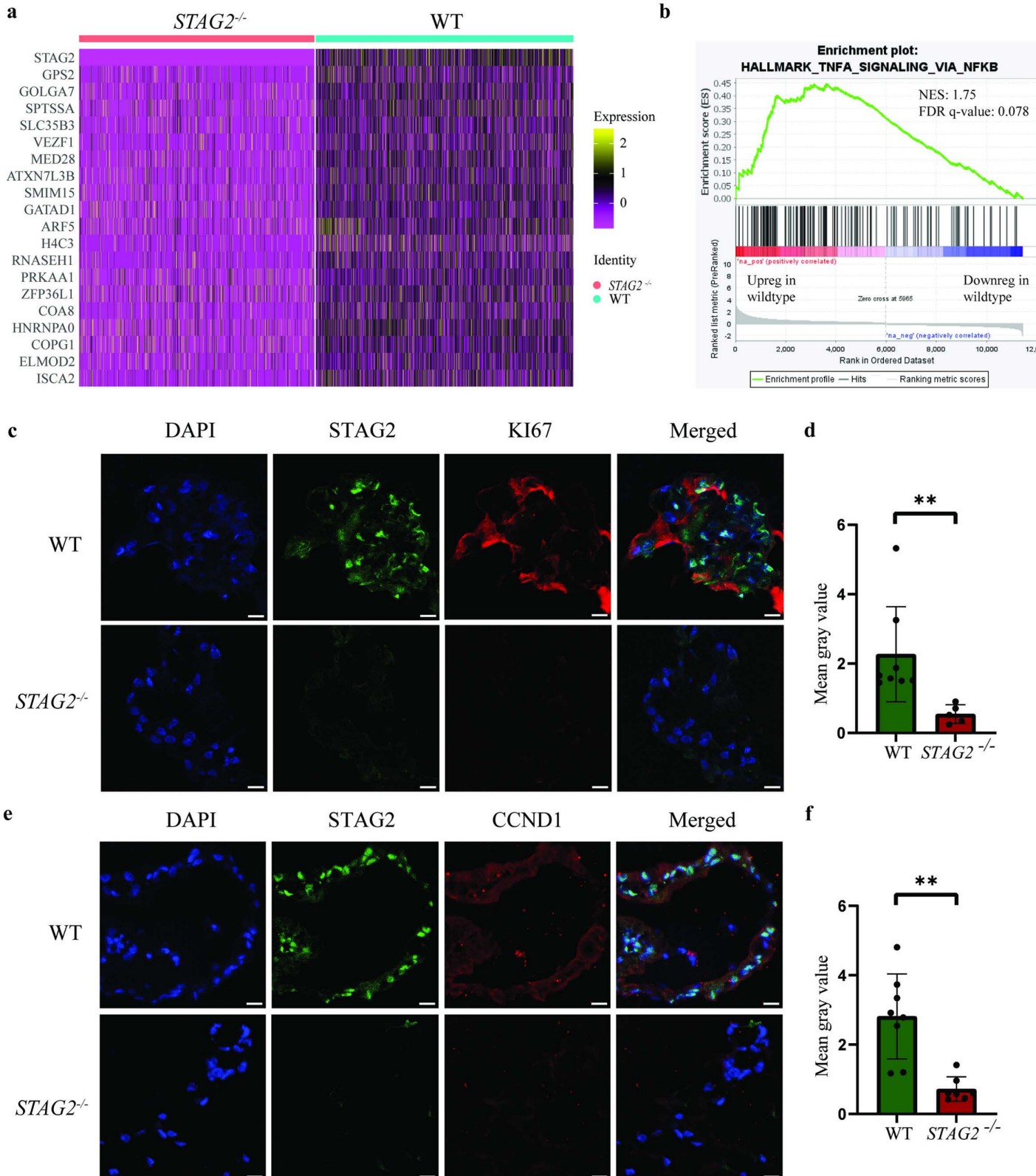

**Fig 4. Analysis of differential expression of genes and proteins between *STAG2* wildtype and mutant populations.** **a.** Heatmap depicting the top 20 differentially expressed genes between *STAG2* wildtype and mutant populations. **b.** Using gene set enrichment analysis (GSEA), only the

"HALLMARK_TNFA_SIGNALING_VIA_NFKB" gene set was upregulated in wildtype compared to mutant (Normalised enrichment score: 1.75, False discovery rate *q*-value: 0.078). **c.** On immunofluorescence staining, we observed increased proliferation of wildtype organoids relative to *STAG2* mutants when stained with anti-KI67 antibody. **d.** Fluorescent intensity of anti-KI67 antibody was statistically significantly upregulated in co-cultured wildtype organoids relative to *STAG2* mutant organoids. **e.** We also observed increased tumorigenicity of wildtype organoids relative to *STAG2* mutants when stained with anti-CCND1 antibody. **f.** Fluorescent intensity of anti-KI67 antibody was statistically significantly upregulated in co-cultured wildtype organoids relative to *STAG2* mutant organoids. Scale bars are 50μm. All experiments were performed with N = 3 biological replicates.

"HALLMARK_KRAS_SIGNALING_UP" (Fig 5f), and "HALLMARK_P53_PATHWAY" (Fig 5g). Notably, Cluster 3, which predominantly comprises of cells which were *STAG2* mutant, none of the Hallmark gene sets were observed to have been upregulated or downregulated at FDR < 25%. Complete GSEA analysis for individual clusters can be found in S1 Table.

Findings on scRNAseq were validated using immunofluorescence staining. In line with upregulated "Hallmark" gene sets, we performed immunofluorescence staining using anti-TNFα (Fig 6a and 6b), anti-KRAS (Fig 6c and 6d) and anti-p53 (Fig 6e and 6f) antibodies. We observed an upregulation of fluorescent intensity in co-cultured wildtype organoids compared with *STAG2* mutant organoids (TNFα: *p* = 0.0031; KRAS: *p* = 0.0021; p53: *p* = 0.0004). Taken together, these observations confirm that oncogenic pathways are upregulated in co-cultured wildtype cells while oncogenic pathways in *STAG2* mutant cells themselves remain relatively quiescent.

In addition, we considered the possibility that co-cultured *STAG2* mutant and wildtype populations would uncover differences in colonic epithelial cell subtypes. Given Cluster 3 had a greater proportion of *STAG2* mutant cells, we sought to compare the composition of colonic epithelial cell subtypes between Cluster 3 and other Clusters. We analysed the expression of markers for the different colonic epithelial cell subtypes. We observed minimal differences in the composition of stem cells with *LGR5* and *PTK7* expression (Fig 7a and 7b), transit amplifying cells with *ZNF277* (Fig 7c), goblet cells with *MUC2* (Fig 7d), and Paneth cells with *LYZ* (Fig 7e), and enterocytes with *KRT20* (Fig 7f).

## Discussion

Although *STAG2* mutations are driver mutations in the normal colon [4], the paucity of *STAG2* mutations in CRC raises questions about the evolutionary trajectory of colon cells possessing this mutation during tumour initiation. Recent publications which performed deep sequencing (up to 100X) to uncover the mutational landscape in large numbers of CRC have failed to identify *STAG2* as a putative driver gene [22,23], suggesting the possibility that *STAG2* mutant cells may instead have a cell interactive effect with neighbouring wildtype cells at an early evolutionary stage. Here, we co-cultured *STAG2* mutant and wildtype cells and revealed subtle transcriptional differences between co-cultured populations. Particularly, we observed an upregulation of oncogenic pathways in neighbouring wildtype cells. Using GSEA, we demonstrated a significant upregulation in genes associated with TNFα signaling. Strikingly, GSEA of scRNAseq clusters which were predominantly comprised of wildtype cells confirmed the upregulation of TNFα signaling, while clusters where wildtype cells were less abundant did not show this. TNFα signaling has been shown to be a critical mediator of CRC progression [24], and suppression of TNFα signaling has conversely induced regression of CRC [25]. The association between upregulation of TNFα signaling gene set and upregulation of the GPS2 gene in wildtype cells is especially intriguing. GPS2 is a known regulator of TNFα signaling by inhibiting downstream JNK activity independent of the NFκB activity [26]. We propose the possibility that upregulated TNFα signaling may induce a homeostatic increase in GPS2 activity to reduce TNFα signaling.

Another pathway potentially upregulated in neighbouring wildtype cells relates to our observed upregulation of *HNRNPA0* activity. The downregulation of *HNRNPA0* via dephosphorylation in colorectal cancer is associated with increased cellular apoptotic events and decreased tumour growth. Conversely, increased *HNRNPA0* activity stabilises chromosomes along the equatorial plane during mitosis and results in tumour progression [27].

Our findings stand in contrast to recent discoveries which have suggested that driver mutations in normal tissue may stave off the onset of carcinogenesis, as in the case of *NOTCH1* in the oesophagus [3] and *FBXW7* in the colon [5]. Our

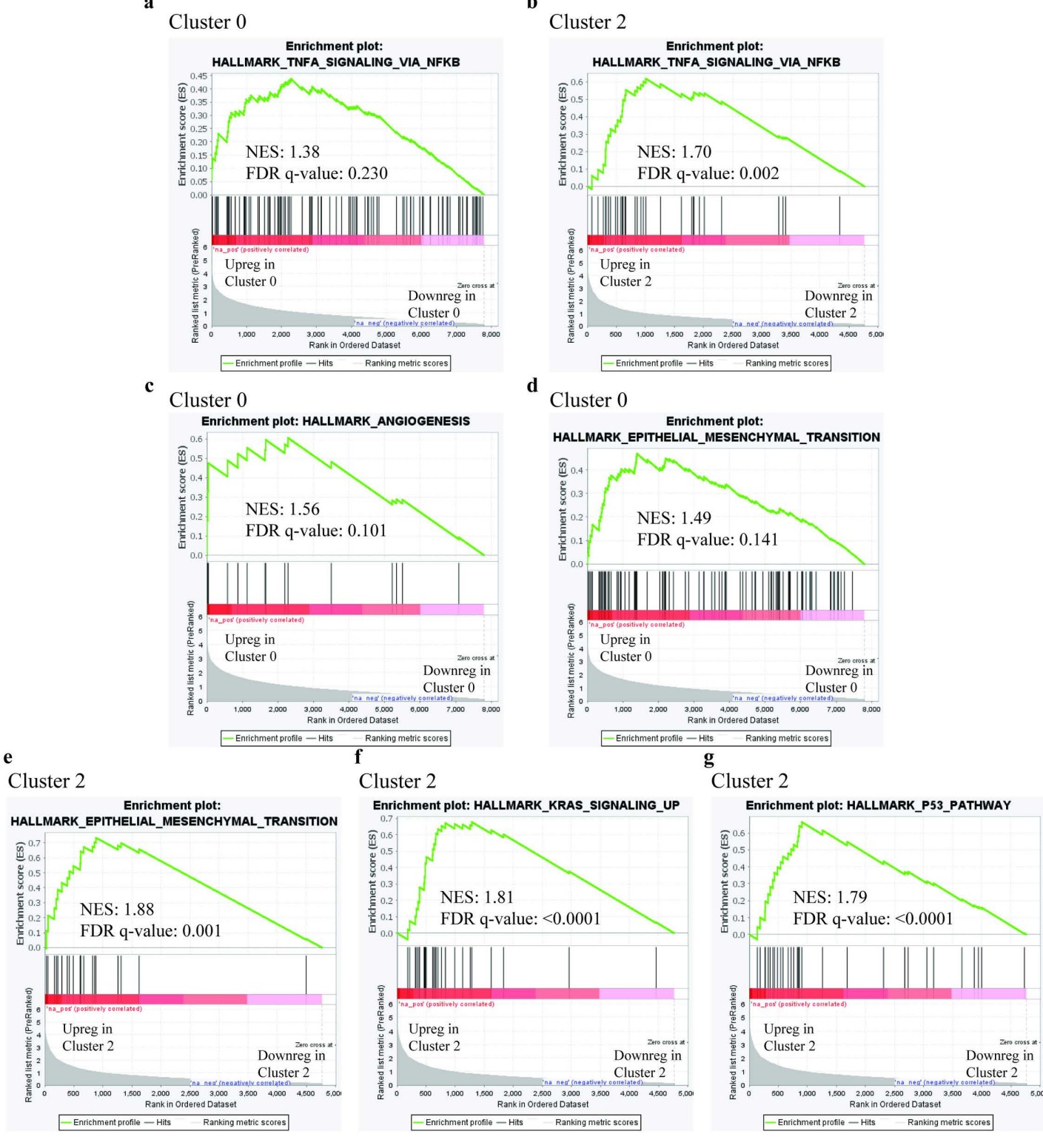

**Fig 5. In depth analysis of transcriptional changes within each cluster suggested an oncogenic effect on co-cultured *STAG2* wildtype cells.** Clusters 0 and 2 were comprised predominantly by wildtype cells. An upregulation of the "HALLMARK_TNFA_SIGNALING_VIA_NFKB" gene set in both Clusters 0 (a) and 2 (b) was observed. In Cluster 0, there was additionally an upregulation of "HALLMARK_ANGIOGENESIS" (c) and

"HALLMARK_EPITHELIAL_MESENCHYMAL_TRANSITION" **(d)**. In Cluster 2, there was additionally an upregulation of "HALLMARK_EPITHELIAL_MESENCHYMAL_TRANSITION" **(e)**, "HALLMARK_KRAS_SIGNALING_UP" **(f)**, and "HALLMARK_P53_PATHWAY" **(g)**.

results suggest that mutations in the normal colon could instead induce the upregulation of cancer-associated pathways via indirect cell interactive mechanisms by altering the expression of genes (Fig 8). This finding has potentially important implications with respect to tumour initiation. Our findings suggest that events which trigger tumour initiation may occur when cells appear phenotypically normal. Moreover, these alterations in gene expression trigger well-known cancer associated pathways, possibly providing the initial steps in carcinogenesis. Finally, our findings demonstrate the potential for a novel mechanism in which CRC oncogenesis may occur, and raises the possibility of further therapeutic targeting. Further research probing molecular mechanisms in which colon cells with *STAG2* mutations upregulate the activity of such pathways in neighbouring wildtype cells will need to be performed.

## Methods

### Human material for organoid cultures

Ethics approval for the retrieval of normal human colon from surgical specimens was accorded by the National Healthcare Group Domain Specific Review Board (Ref: 2023/00274). All patients gave informed written consent. Recruitment of participants commenced on 01/10/2023 and was completed on 31/09/2024.

### Organoid culture

Colonic tissue was obtained from adult patients who were undergoing surgery for both benign and malignant colorectal disease. In cases where patients underwent surgery for colorectal cancer, harvested colonic tissue was situated at least 5 cm from the tumour edge. A 1 x 1 cm piece of colon appearing phenotypically normal was resected. Organoids were derived based on our previously published protocol [12]. Organoid culture growth media consists of advanced DMEM/F12 (Gibco), and supplemented with penicillin/streptomycin (Gibco), 10mM HEPES buffer solution (Gibco), 2mM GlutaMAX (Gibco), 50% Wnt3a conditioned medium (produced from ATCC CRL-2647 cell line), 25% R-spondin conditioned medium (produced from Cultrex HA-R-Spondin1-Fc 293T cells), 1X B-27 plus supplement (Gibco), 10μM SB 202190 (Tocris), 0.5μM SB 431542 (Tocris), 1μM prostaglandin E2 (Tocris), 50ng/ml Noggin (Peprotech), 336 50ng/ml EGF (Peprotech), 10mM nicotinamide (Sigma-Aldrich), and 1.25mM N-acetylcysteine (Sigma-Aldrich).

For selection of *APC* mutants, Wnt3a and R-spondin conditioned media were withdrawn from the growth media for at least 4 weeks.

### CRISPR/Cas9 gene editing of wildtype organoids

Guide RNAs (gRNAs) were purchased from EditCo for both the *STAG2* and the *APC* gene. For *STAG2*, a multiguide approach comprising of 3 different sgRNAs was used (Table 2). We used an electroporation approach which has been described in detail previously [12]. Briefly, cystic, early-passage organoids were dissociated into single-cell suspension using TrypLe (Gibco). A ribonucleoprotein (RNP) complex was generated by mixing 25 μM of Cas9 enzyme (Sigma-Aldrich), and 100 μM of the sgRNA to generate a molar ratio of 1:1 after mixing. Using a Neon Transfection System (ThermoFisher), cells were electroporated with the RNP complex at voltage 1300 V, width 20 ms and 2 pulses. Cells were then incubated for one week. After one week, transfection efficacy was checked by Sanger sequencing (1st Base), using an amplicon generated with the forward and reverse primers during polymerase chain reaction (PCR) (Table 2).

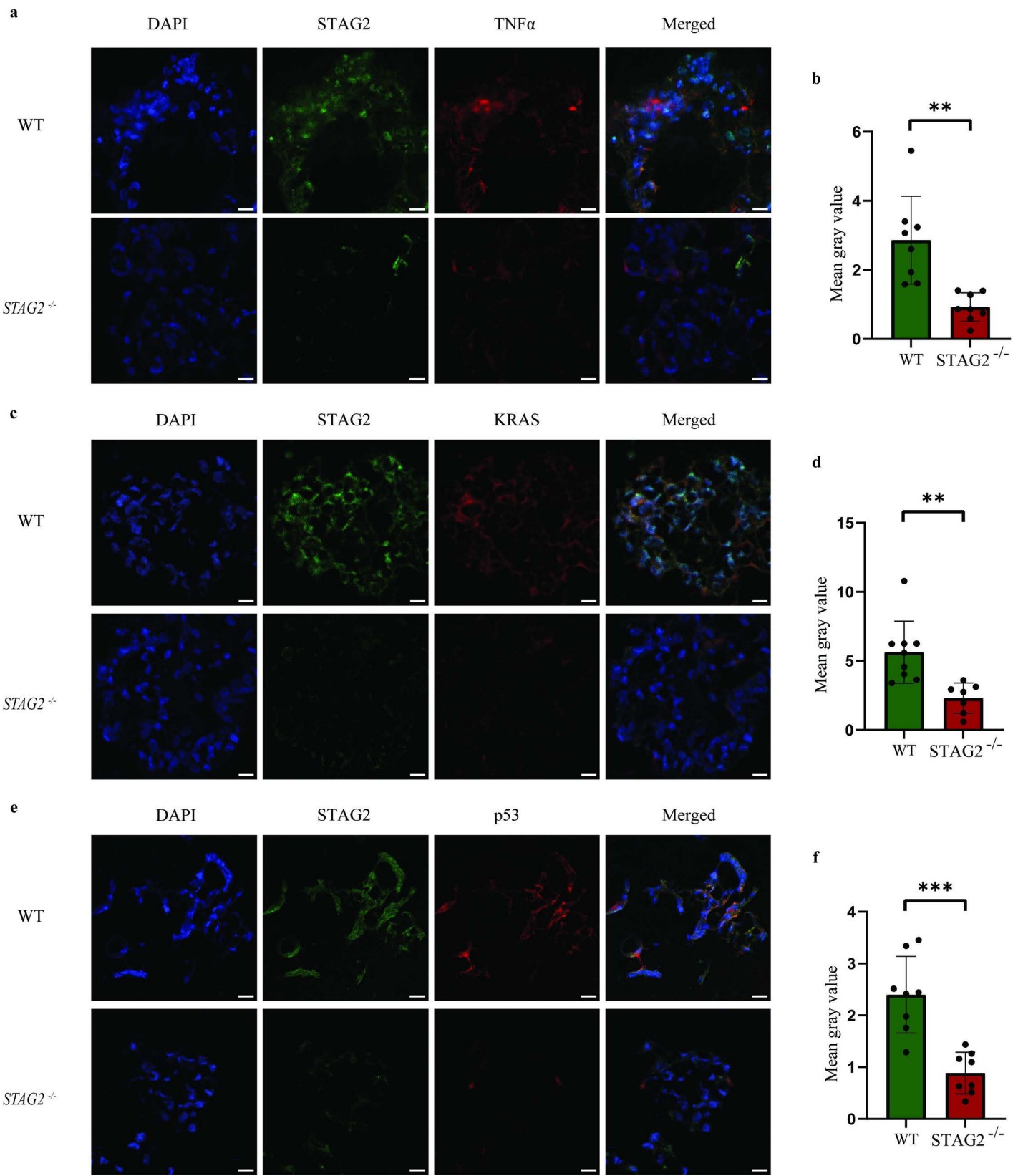

**Fig 6. Validation of scRNAseq findings was performed using immunofluorescent staining against target proteins.** Fluorescent intensity of anti-TNFα antibody **(a, b)**, anti-KRAS antibody (c, d) and anti-p53 antibody (e, f) was statistically significantly upregulated in co-cultured wildtype organoids relative *STAG2* mutant organoids. Scale bars are 50µm. All experiments were performed with N = 3 biological replicates.

## Protein lysates and western blot

Proteins were harvested using RIPA (ThermoFisher) with the addition of protease and phosphatase inhibitor cocktails (Sigma-Aldrich). Quantification of protein concentration was performed using BCA assay (ThermoFisher). For the western blot, 30 µg of protein was used for each lane. Proteins were separated using a NuPage 4–12% Bis-Tris gel (Invitrogen) and transferred onto a 0.45 µm nitrocellulose membrane. The membrane was blocked with 5% low-fat milk for 1 h before overnight incubation with primary antibodies at 4 °C. Rabbit anti-human STAG2 antibody (1:2500, ab201451, Abcam, USA), rabbit anti-human RAD21 antibody (1:2500, ab, Abcam, USA), rabbit anti-human STAG1 antibody (1:2500, Abcam, USA), and rabbit anti-human GAPDH antibody (1:2500, ab9485, Abcam, USA) was used. The membrane was then incubated with goat anti-rabbit antibody (1:5000, ab6721, Abcam, USA) at room temperature for 1 h. The membranes were then imaged with a ChemiDoc XRS+ system (Bio-Rad).

## Immunofluorescence and imaging

Media was removed from the wells of organoids, and the BME was broken down by incubating with TrypLe (Gibco) at 37°C for 10 min. Organoids were then palleted by centrifugation at 400g for 5 min. The organoid pellet was then resuspended in Optimal Cutting Temperature (OCT) and frozen through placement in liquid nitrogen. OCT cryostat sectioning was performed to a thickness of 10µm and mounted onto slides.

For immunofluorescent experiments in Fig 1, slides were first blocked with 5% goat serum in PBS. Thereafter, blocking solution was removed, and slides were incubated with rabbit anti-human STAG2 antibody (1:100, 19837–1-AP, Proteintech, USA) for two hours at room temperature. Slides were then washed in PBS, before the addition of multi-rAb Coralite Plus 488-Goat Anti-rabbit antibody (1:500, RGAR200, Proteintech, USA), Coralite Plus 647-conjugated beta-actin recombinant antibody (1:500, CL647–81115, Proteintech, USA) and DAPI(0.1 µg/ml, D1306, Invitrogen, USA) for one hour at room temperature. Slides were washed again with PBS. Fluorescent imaging was performed with the LSM 880 with Airyscan (Zeiss, Germany).

For immunofluorescent experiments in Figs 4 and 5, unless otherwise stated, steps were similar as above. Primary antibodies used included rabbit anti-human STAG2 antibody (1:100, 19837–1-AP, Proteintech, USA), mouse anti-human KI67 antibody (1:500, 66555–6-Ig, Proteintech, USA), mouse anti-human P53 antibody (1:400, 60283–2-Ig, Proteintech, USA), mouse anti-human CCND1 antibody (1:100, 60186–1-Ig, Proteintech, USA), mouse anti-human TERT antibody (1: 100, MA5−16033, Invitrogen, USA), mouse anti-human KRAS antibody (1:250, 415700, Invitrogen, USA), and mouse anti-human TNFα antibody (1:50, MA5−23720, Invitrogen, USA). Slides were then washed in PBS, before the addition of CoraLite488-conjugated Goat Anti-Rabbit antibody (1:500, SA00013−2, Proteintech, USA), Multi-rAb™ CoraLite® Plus 647-Goat Anti-Mouse antibody (1:500, RGAM005, Proteintech, USA), and DAPI(0.1 µg/ml, D1306, Invitrogen, USA). Fluorescent imaging was performed with the SP8 Lightning (Leica, Germany).

## Differentiation of organoids, imaging and characterisation

To achieve differentiation of organoids, Intesticult Differentiation Media (Stemcell Technologies) with addition of 5µM of DAPT was used as the organoid culture media. Organoids were cultured in differentiation media for 7 days before being imaged using an Olympus LX71 microscope at 10X magnification. For measurement of circularity, the "Circularity" function on ImageJ 1.54g was used [28]. For measurement of crypts/mm2, surface area of the organoid was first calculated on ImageJ. The number of crypts were manually counted, and divided against the surface area.

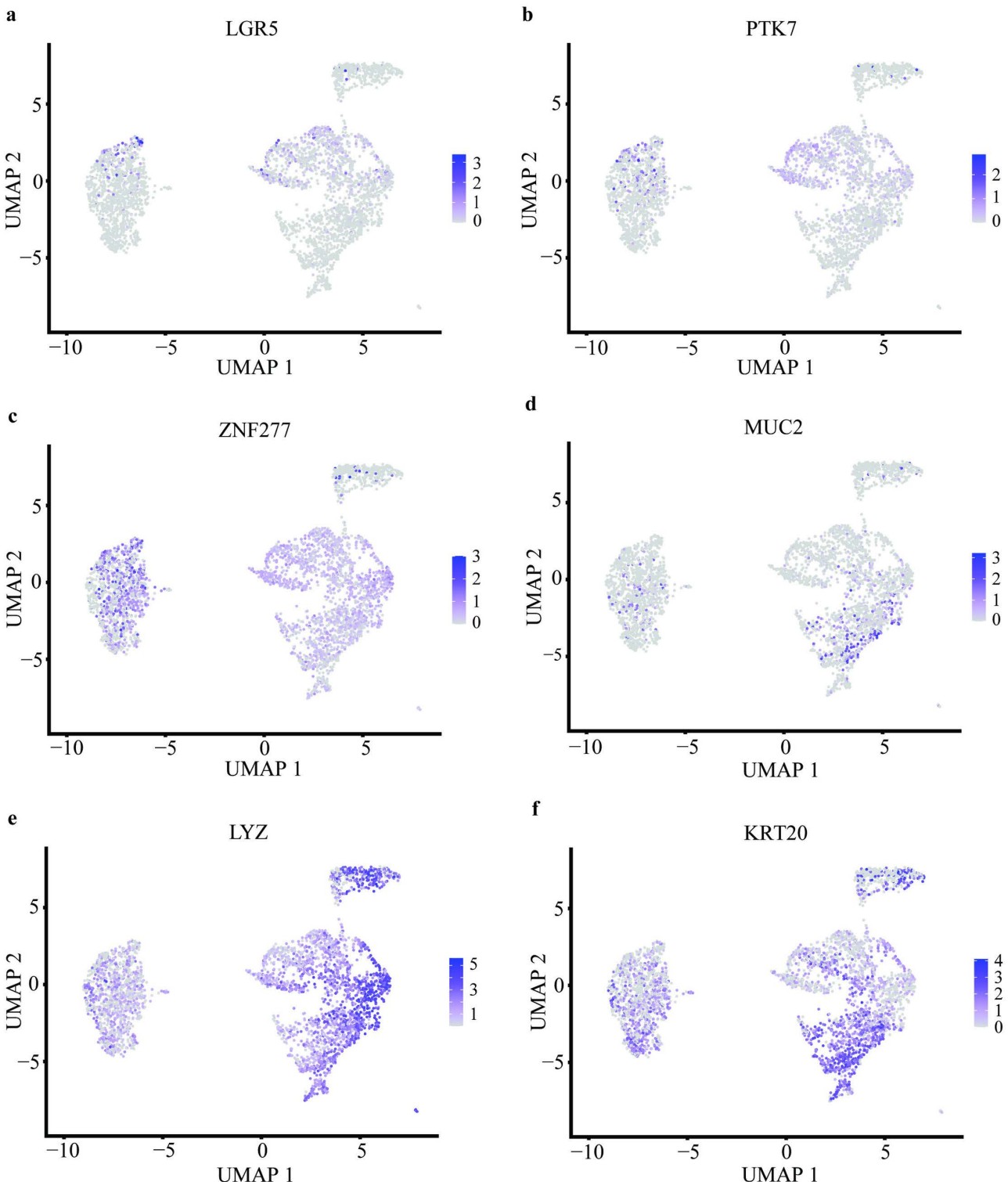

**Fig 7. Minimal differences were observed in cellular subtypes between *STAG2* wildtype and mutant cells.** Comparing between wildtype and mutant populations, there were no differences in the composition of stem cells with *LGR5* (a) and *PTK7* (b) expression, transit amplifying cells with *ZNF277* **(c)**, goblet cells with *MUC2* **(d)**, and Paneth cells with *LYZ* **(e)**, and enterocytes with *KRT20* **(f)**.

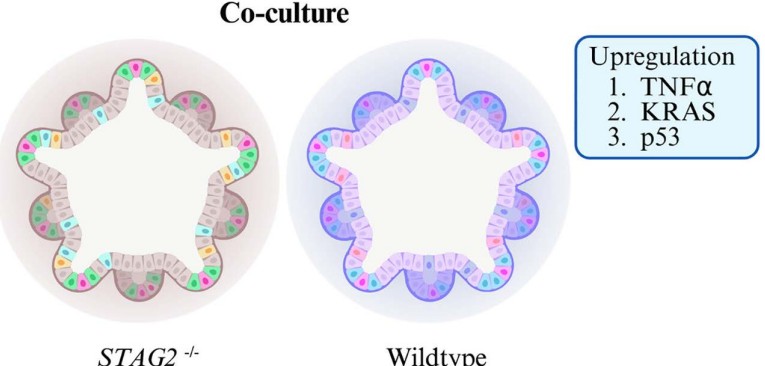

**Co-culture**

Upregulation
1. TNFα
2. KRAS
3. p53

*STAG2 -/-*          Wildtype

**Fig 8. Diagram demonstrating the effect of co-cultured *STAG2* mutant organoids and wildtype organoids.** In co-culture, we observed upregulation of TNFα, KRAS and p53 in wildtype organoids, proposing a cooperative mechanism of early oncogenesis.

**Table 2. gRNA sequences used for CRISPR-Cas9 and associated primer sequences to validate gene knockout on Sanger Sequencing.**

| gRNA sequences for CRISPR-Cas9 | | |
|---|---|---|
| Gene target | sgRNA sequences | |
| *APC* | UCUGUAUAAAUGGCUCAUCG | |
| *STAG2* | ACUUGUAAAAAAGGCAAAAA UCUGGUCCAAACCGAAUGAA UUGUUUGAAGUUGUUAAAAU | |
| Sequencing primers to validate gene knockout | | |
| Gene target | Forward primer | Reverse primer |
| *APC* | ATGCTGCAGTTCAGAGGGTC | TTTTTCTGCCTCTTTCTCTTGGT |
| *STAG2* | GGACACCACAAAGAGGCTGT | ACATCCCAAGAGTTTTCTGATGA |

## Organoid co-culture

An equal number of wells (1:1) of *STAG2* wildtype and mutant organoids were cultured separately as previously described. Media was removed from each well of organoids and incubated on ice in Organoid Harvesting Solution (Cultrex) for 30 min. Thereafter, organoids were collected into a 15 ml centrifuge tube before palleting organoids at 400g for 5 min. Palleted organoids were resuspended in advanced DMEM (Gibco), and *STAG2* wildtype and mutant organoids were mixed together. Organoids were palleted again by centrifugation at 400g for 5 min. Organoids were then resuspended in BME (Cultrex) before before grown in 24-well plates. Organoids grew in co-culture for at least 14 days before being subjected to onward experiments.

## Analysis of fluorescence

Analysis of fluorescence was performed on ImageJ 1.54g using the "Analyze>Measure" function after drawing a rectangle around individual organoids [28]. The "Mean Gray Value" was used to assess the fluorescent intensity of each individual organoid.

## Single-cell RNAseq and analysis

Extraction of total RNA was performed using the RNeasy Mini Kit (Qiagen) as per manufacturer's protocol. The quantity and quality of the RNA was ascertained using the Agilent 2100 Bioanalyzer. A single cell 3' gene expression dual index

library was generated following the manufacturer's instructions (10X Genomics) and post library construction QC was performed using the Agilent Bioanalyzer High Sensitivity chip. Libraries were then sequenced on an Illumina NovaSeq X Plus.

Raw sequencing data was aligned to the human reference genome (GRCh38, v2024-A) using the CellRanger count pipeline (10X Genomics, v8.0.1). The filtered feature-barcode matrices were processed and analyzed using the Seurat package [29]. *STAG2* mutant and wildtype cells were inferred from co-cultured samples by inspecting the abundance of *STAG2* gene expression in feature-barcode matrices. Cell barcodes with a non-zero expression of *STAG2* were manually labelled as wildtype and cell barcodes with a zero expression of STAG2 were manually labelled as mutant. Cell barcodes with >20% mitochondrial reads were removed from feature-barcode matrices. The standard Seurat workflow was then used to normalize, scale and group cells into distinct clusters based on their gene expression profiles. Mitochondrial, ribosomal and long non-protein coding RNA genes were removed prior to running the standard workflow as they were not of primary interest in this study. A Wilcox Rank Sum test was used to (1) identify differentially expressed genes between clusters using the FindAllMarkers function; and (2) identify differentially expressed genes between STAG2 mutant and wildtype cells using the FindMarkers function [29]. Heatmaps were used to (1) visualize differentially expressed genes between clusters; and (2) visualize differentially expressed genes between *STAG2* mutant and wildtype cells using the DoHeatMap function [29]. The uniform manifold approximation and projection (UMAP) ordination method was used to visualize relationships between cell clusters and *STAG2* mutant and wild type cells using the DimPlot function [29]. The abundance of genes associated with intestinal stem cells (*LGR5*, *PTK7*), goblet cells (*MUC2*), paneth cells (*LYZ*), transit amplifying cells (*ZNF277*) and epithelial cells (*KRT20*) were visualized using the FeaturePlot function [29].

## Supporting information

**S1 Table. Complete results of gene set enrichment analysis (GSEA) between *STAG2* wildtype and mutant cells within each cluster from scRNAseq.**
(XLSX)

**S2 Table. Differential expression of genes between wildtype and *STAG2* mutant clusters.**
(XLSX)

**S1 Raw Images. Uncut western blot images.**
(TIF)

**S1 Fig. Heatmap depicting differentially expressed genes among the four clusters.**
(TIF)

## Author contributions

**Conceptualization:** Dedrick Chan.

**Data curation:** Dedrick Chan.

**Formal analysis:** Dedrick Chan, Christopher James Dean.

**Funding acquisition:** Dedrick Chan.

**Investigation:** Dedrick Chan, Wei Ni Yew, Christopher James Dean.

**Methodology:** Dedrick Chan, Christopher James Dean.

**Project administration:** Dedrick Chan.

**Resources:** Dedrick Chan.

**Supervision:** Dedrick Chan.

**Validation:** Dedrick Chan, Christopher James Dean.

**Writing – original draft:** Dedrick Chan.

**Writing – review & editing:** Dedrick Chan, Wei Ni Yew, Christopher James Dean.

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
