## [Decision Letter · Decision Letter 0]

28 Apr 2025

PONE-D-25-14812STAG2 mutations in the normal colon induce upregulation of oncogenic pathways in neighbouring wildtype cellsPLOS ONE

Dear Dr. Chan,

Thank you for submitting your manuscript to PLOS ONE. After careful consideration, we feel that it has merit but does not fully meet PLOS ONE’s publication criteria as it currently stands. Therefore, we invite you to submit a revised version of the manuscript that addresses the points raised during the review process.

We look forward to receiving your revised manuscript.

Kind regards,

Xinjun Lu

Academic Editor

PLOS ONE

“Dedrick Kok Hong Chan was supported by an N2CR-CSDU CS Pilot Grant as well as an NUHS Clinician Scientist Program grant (NCSP2.0/2023/NUHS/DCKH).”

5. Please include your tables as part of your main manuscript and remove the individual files. Please note that supplementary tables (should remain/ be uploaded) as separate "supporting information" files

Reviewers' comments:

Reviewer's Responses to Questions

**Comments to the Author**

1. Is the manuscript technically sound, and do the data support the conclusions?

Reviewer #1: Yes

Reviewer #2: Yes

2. Has the statistical analysis been performed appropriately and rigorously? 

Reviewer #1: I Don't Know

Reviewer #2: Yes

3. Have the authors made all data underlying the findings in their manuscript fully available?

Reviewer #1: Yes

Reviewer #2: Yes

4. Is the manuscript presented in an intelligible fashion and written in standard English?

Reviewer #1: Yes

Reviewer #2: Yes

5. Review Comments to the Author

Reviewer #1: 1. The hypothesis that STAG2 mutations affect neighboring wildtype cells via non-cell-autonomous mechanisms is interesting, and the approach is suitable.

2. The connection to CRC is tenuous, and will necessitate functional verification (e.g., tumorigenicity assays).

3. Inclusion of APC mutants as positive controls is commendable. However, co-culture conditions and proportions of mutant:wildtype cells should be quantitatively detailed.

4. Current interpretation is speculative; prioritize elucidating definitive molecular mechanisms.

5. Provide a summary schematic describing the proposed mechanism in this study to enhance the understanding.

Reviewer #2: The authors have revealed that STAG2 mutations in normal colon organoids upregulates the expression of key oncogenic pathways in neighboring cells not bearing STAG3 mutations. This is a very interesting and timely paper. I have no comments on this well-written, intriguing paper other than further information is needed in the Discussion describing potential mechanisms (exosomes for example) and the fact that it is known and numerous papers in the literature demonstrating that mutations in stroma, namely fibroblasts, can activate pro-proliferative and oncogenic pathways in neighboring epithelia.

6. PLOS authors have the option to publish the peer review history of their article (what does this mean? ). If published, this will include your full peer review and any attached files.

**Do you want your identity to be public for this peer review?** For information about this choice, including consent withdrawal, please see our Privacy Policy .

Reviewer #1: No

Reviewer #2: No

---

## [Author Response · Author response to Decision Letter 1]

15 Aug 2025

Dear Editors and Reviewers,

On behalf of the authors, we would like to thank the editors and reviewers for the opportunity to revise our manuscript. The long interim since review was because we took the comments seriously, and performed further experiments to establish the claims in our work. We hope that the editors and reviewers will find these revisions satisfactory.

Please find a point by point reply to the comments made.

Journal requirements

1. We have endeavoured to adhere to PLOS ONE’s styles.

2. Funding statement should read “ Dedrick Kok Hong Chan was supported by an N2CR-CSDU CS Pilot Grant as well as an NUHS Clinician Scientist Program grant (NCSP2.0/2023/NUHS/DCKH).” I have removed the remarks in the Acknowledgement section.

3. Indeed, all sequence reads have already been uploaded in the public domain. I have included the Sequence Read Archive accession number and link in the paper.

4. All western blot uncropped images are found in S1 Fig.

5. We have included tables in the main manuscript file. We also moved Table 2 to Supplementary information.

Reviewer 1

1. We are appreciative that the reviewer felt this was an appropriate approach to investigate our question.

2. We performed additional experiments to address this. In co-cultured organoids, we stained for Ki67 and Cyclin D1, markers of tumorigenicity. We found an upregulation of both in the co-cultured wildtype organoids relative to the STAG2 mutant organoids. These results are presented.

3. We have included further information about co-culture conditions in the Methods section.

4. We appreciate this comment. To elucidate molecular mechanisms, we performed further immunofluorescent experiments to demonstrate that, in concordance with the sequencing results, upregulation of p53, KRAS and TNFα. Further molecular characterisation will form a future study in which a detailed analysis will be performed. We have included these new results as a new Figure 6.

5. We have included a Figure 8 to demonstrate the findings of this paper in a picture.

Reviewer 2

1. The authors appreciate the highly encouraging comments of this reviewer. Particular, we thank the reviewer for noting that our paper was “very interesting and timely”.

Again, the authors thank the editors and reviewers for this opportunity to revise our paper.

With gratitude,

Dr Dedrick Chan

---

## [Decision Letter · Decision Letter 1]

1 Sep 2025

STAG2 mutations in the normal colon induce upregulation of oncogenic pathways in neighbouring wildtype cells

PONE-D-25-14812R1

Dear Dr. Chan,

We’re pleased to inform you that your manuscript has been judged scientifically suitable for publication and will be formally accepted for publication once it meets all outstanding technical requirements.

Kind regards,

Xinjun Lu

Academic Editor

PLOS ONE

Additional Editor Comments (optional):

Reviewer #1:

Reviewer #2:

Reviewers' comments:

Reviewer's Responses to Questions

**Comments to the Author**

1. If the authors have adequately addressed your comments raised in a previous round of review and you feel that this manuscript is now acceptable for publication, you may indicate that here to bypass the “Comments to the Author” section, enter your conflict of interest statement in the “Confidential to Editor” section, and submit your "Accept" recommendation.

Reviewer #1: All comments have been addressed

Reviewer #2: All comments have been addressed

2. Is the manuscript technically sound, and do the data support the conclusions?

Reviewer #1: Yes

Reviewer #2: Yes

3. Has the statistical analysis been performed appropriately and rigorously? 

Reviewer #1: I Don't Know

Reviewer #2: Yes

4. Have the authors made all data underlying the findings in their manuscript fully available?

Reviewer #1: Yes

Reviewer #2: Yes

5. Is the manuscript presented in an intelligible fashion and written in standard English?

Reviewer #1: Yes

Reviewer #2: Yes

6. Review Comments to the Author

Reviewer #1: The authors have satisfactorily addressed the previous reviewer comments, and the manuscript is technically sound with data that support the conclusions.

Reviewer #2: (No Response)

7. PLOS authors have the option to publish the peer review history of their article (what does this mean? ). If published, this will include your full peer review and any attached files.

**Do you want your identity to be public for this peer review?** For information about this choice, including consent withdrawal, please see our Privacy Policy .

Reviewer #1: No

Reviewer #2: No

---

## [Editor Report · Acceptance letter]

PONE-D-25-14812R1

PLOS ONE

Dear Dr. Chan,

I'm pleased to inform you that your manuscript has been deemed suitable for publication in PLOS ONE. Congratulations! Your manuscript is now being handed over to our production team.

Kind regards,

on behalf of

Dr. Xinjun Lu

Academic Editor

PLOS ONE